# Leveraging MLLMs for Zero-Shot Action Recognition: Concise, Discriminative and anti-Hallucination Prompting

## Abstract

Leveraging the capabilities of large language models (LLMs), multi-modal LLMs (MLLMs) show great promise for zero-shot action recognition (ZSAR). However, current MLLM-based approaches often struggle to accurately locate the right action from many, largely due to issues such as lengthy, vague prompts and hallucinated outputs. In this paper, we introduce CDantiHalP (**C**oncise, **D**iscriminative and **anti-Hal**lucination **P**rompting), a novel LLM-driven approach to enhance MLLM performance in ZSAR. CDantiHalP is a training-free, post-refinement method designed to improve recognition accuracy for any baseline model. It consists of two core components: (1) concise, discriminative prompting to effectively distinguish confused action pairs, and (2) logic-contradictory hallucination detection (LogCHalD) to identify and mitigate hallucinations. Rather than relying on MLLMs to select from a broad set of labels, CDantiHalP leverages their strength in pairwise comparison of specific concepts. The use of concise-discriminative prompts highlights the distinguishing features between confused actions, guiding MLLMs to focus on critical differences while remaining alert to potential hallucinations. The LogCHalD framework further enhances response reliability by using a logic-contradictory strategy to detect hallucinated responses for each confused action pair. During inference, CDantiHalP assesses hallucination risk and emphasizes consistency across MLLM outputs to mitigate the impact of hallucinations. Extensive experiments demonstrate that CDantiHalP achieves state-of-the-art performance on various ZSAR datasets.

## 1 Introduction

The rapid development of zero-shot action recognition (ZSAR) has been facilitated by the emergence of vision-language pre-training models, such as CLIP Radford et al. (2021) as well as the LLM-driven multi-modal large language models (MLLMs). Compared to the CLIP-based methods Pan et al. (2022); Yang et al. (2023); Ju et al. (2022); Wu et al. (2023b), MLLMs Wang et al. (2024c); Maaz et al. (2024); Lin et al. (2024); Zhang et al. (2024); Cheng et al. (2024a) have shown more impressive results due to the powerful capabilities of LLMs in understanding video content. However, effectively leveraging MLLMs for ZSAR without requiring additional data for training or fine-tuning, i.e., achieving training-free learning, remains an open research challenge.

To adapt MLLMs normally trained on open-ended QA datasets to closed-set ZSAR tasks, a simple solution is to present all predefined labels and let the model choose. However, this often degrades performance due to long, vague token sequences Liu et al. (2024). A stronger alternative is a two-stage pipeline: CLIP-based pre-classification followed by MLLM inference over the top-k candidates Liu et al. (2024). Another approach Hanu et al. (2023) generates video captions for LLM-based label inference, though this risks errors from hallucinated or irrelevant descriptions.

In this paper, we explore the potential of MLLMs for ZSAR with a focus on the aforementioned two challenges. Since the typical MLLM conversation pipeline involves inputting a prompt alongside the video to guide the generation of responses that directly or indirectly identify the action, As illustrated in Figure 1, this raises two key concerns. (1) [***Prompt design***]: Effective prompts are critical for unlocking the full potential of MLLMs. As shown by Liu et al. (2024), MLLMs struggle

Figure 1: Intuition behind prompt design and response trustworthiness. The top example illustrates the necessity of a more discriminative prompt to differentiate between two easily confused actions. The right example indicates that MLLMs may tend to favor a positive response, particularly when the frame is of low quality, highlighting the need for effective hallucination detection.

with long token sequences, particularly those nuanced or easily confused action labels, leading to degraded performance. This highlights the need for **concise** prompts to avoid overwhelming the model. Additionally, vague prompts (e.g., "describe the scene or action") lead to broad, unfocused responses, complicating the recognition of specific actions. **Discriminative** prompts that guide the MLLM to focus on key action details are thus critical for accurate ZSAR. (2) [*Response trustworthiness*]: Like LLMs, MLLMs are prone to **hallucinations** Bai et al. (2024); Xu et al. (2024b), where responses may be nonfactual or inconsistent with visual input. This often occurs due to the model's prior knowledge overshadowing actual video details, and biases in training data that favor positive ("yes") responses. Hallucinations typically manifest as errors in object existence, attributes, or relationships Bai et al. (2024), all of which can severely compromise action recognition. This underscores the importance of mechanisms to detect and mitigate hallucinations.

To address the above challenges of ZSAR with MLLMs, we propose a **C**oncise, **D**iscriminative and **anti-Hal**lucination **P**rompting (**CDantiHalP**), a novel approach driven by LLMs. CDantiHalP enhances MLLM performance in distinguishing confused action pairs while mitigating hallucinations. It mainly comprises (1) concise, discriminative prompting for confused action pair distinction and (2) a logic-contradictory hallucination detection (LogCHalD) framework. Notably, CDantiHalP is a training-free method that does not require any additional video data from the target dataset and can be applied as an off-the-shelf solution to refine pre-obtained recognition results from any model.

The design of our concise-discriminative prompt ($\mathbf{Prompt}_{CD}$) addresses the limitations of lengthy and vague prompts, which often hinder MLLM interaction with video content and result in unfocused responses. Rather than requiring the MLLM to choose from a broad set of labels, CDantiHalP leverages the MLLM's strengths in comparing specific pairs of concepts within the video context. This is motivated by the observation that top-1 predictions frequently co-occur with a fixed set of top-2 labels in action recognition models. These co-occurring labels are often semantically or visually similar, which complicates distinction, as demonstrated by a drop in top-1 accuracy from 88.7% (top-2 accuracy) to 76.1% on UCF101 using TEAR Bosetti et al. (2024). We address this by building a confused pair dictionary through co-occurrence analysis and designing $\mathbf{Prompt}_{CD}$ to explicitly contrast these pairs. Prompts are automatically generated by an LLM under system instructions that highlight key distinctions and mitigate hallucinations. Additionally, our method includes an adaptive mechanism to recognize whether an action pair has static or dynamic attributes, thereby determining whether to use frame or video input for optimal interaction with MLLM.

To ensure the reliability of the MLLM's responses obtained with $\mathbf{Prompt}_{CD}$, especially in cases of low-quality video inputs, we introduce LogCHalD, which uses a logic-contradictory prompt ($\mathbf{Prompt}_{LC}$) to challenge the initial response. Inspired by Wu et al. (2024), $\mathbf{Prompt}_{LC}$ is automatically crafted to consider both the confused action pair and its corresponding $\mathbf{Prompt}_{CD}$. By comparing the initial discriminative response with the logic-contradictory response, we can detect inconsistencies indicative of hallucinations. This verification uses an LLM with a logic verification prompt, which systematically checks for contradictions between the two responses.

Finally, CDantiHalP generates a concise-discriminative prompt and assesses the hallucination risk for each confused action pair. All operations within CDantiHalP are one-time processes, making it efficient for practical use. During inference, we introduce a recognition refinement method that further incorporates hallucination mitigation. This method uses an additional LLM prompt to accurately identify the correct action label by emphasizing consistency across responses from different frames, thereby minimizing the influence of hallucinations and boosting recognition accuracy.

Our main contributions are summarized as follows:

- We propose the concise, discriminative prompting method to enhance MLLM performance in ZSAR by emphasizing key distinctions between confused action pairs. This novel approach guides the MLLM to focus on relevant features while being aware of potential hallucinations, leading to more accurate action recognition.

- We introduce the LogCHalD framework to address the challenge of hallucination detection. By utilizing logic-contradictory prompts, this framework identifies inconsistencies in the MLLM's outputs, allowing for the detection and mitigation of hallucinations.

- CDantiHalP is a training-free, post-refinement method that efficiently refines recognition results from any baseline model. Experimental results on various benchmarks also demonstrate its efficacy.

## 2 RELATED WORKS

**CLIP for ZSAR**. Existing approachesfully utilize CLIP's· Radford et al. (2021) strong alignment capability between the visual and the textual contents. ActionCLIP Wang et al. (2023b) is a pioneering work to introduce CLIP into video recognition, fine-tuning the CLIP model and applying additional temporal layers to model motion. X-CLIP Ni et al. (2022) proposes frame-level temporal attention to reduce computation. Another common practice is to freeze the pre-trained model parameters and introduce extra parameters for training. ST-Adapter Pan et al. (2022) and AIM Yang et al. (2023) employ lightweight adapters to transfer knowledge from the image to the video domain. Similarly, EZ-CLIP Ahmad et al. (2023), Vita-CLIP Wasim et al. (2023), VideoPrompt Ju et al. (2022) and BIKE Wu et al. (2023b) leverage learnable prompts to improve recognition performance. MAXI Lin et al. (2023) and OST Chen et al. (2024) explore the text descriptions for action labels. These methods have achieved impressive results, but all require fine-tuning on video data, leading to significant costs in data collection and training. Additionally, TEAR Bosetti et al. (2024) is a training-free method that simply enriches textual descriptions for action labels without fine-tuning the CLIP encoders.

**MLLMs for ZSAR**. MLLMs Li et al. (2025); Liu et al. (2023b); Cheng et al. (2024a); Zhang et al. (2024); Lin et al. (2024); Wang et al. (2024c); Maaz et al. (2024); Wang et al. (2023a); Cheng et al. (2024b) expand LLM architectures by incorporating more additional modalities. These MLLMs serve as foundations for various visual understanding tasks like captioning Tang et al. (2024); Xu et al. (2024a), and question answering (QA) and reasoning Min et al. (2024); Wang et al. (2024d); Guo et al. (2022); Xie et al. (2025). For the studied ZSAR, the most straightforward approach involves feeding the question and all action labels into an MLLM to predict one. However, inputting all possible labels at once can result in performance degradation due to the fine-grained and complex categories. An improved method is the "retrieving and reranking" Liu et al. (2024), which first retrieves the top-k action candidates and then reranks them by an MLLM. While this reduces prompt length and complexity, it does not significantly enhance the model's interactive understanding of video content. Another approach Hanu et al. (2023) leverages MLLMs as visual captioners to generate language descriptions, which are then used by LLMs to infer the action label. However, given the diverse and complex nature of video content, it can result in hallucinated or irrelevant descriptions that do not accurately capture the action label. In contrast, our CDantiHalP effectively shifts the challenge of lengthy and vague prompting to the more manageable task of distinguishing between action pairs, utilizing concise and discriminative prompts that align with MLLMs' strengths.

**Hallucination**. Many recent solutions have focused on detecting and mitigating hallucinations from data, models, training, and inference. The works Liu et al. (2023a); Wang et al. (2024a); Yu et al. (2024); Yue et al. (2024) concentrate on improving data quality by introducing negative and counterfactual data and eliminating noise or errors in existing datasets. In Liu et al. (2023a); Zhai et al. (2023), scaling the resolution of the image encoder is observed to reduce the degree of hallucination with the cost of higher computational demand. Another group of works addresses hallucination in a more explainable and reliable way. They first detect suspicious information in MLLMs' outputs with external tools Yin et al. (2023) or specifically designed mechanisms Wu et al. (2024) and then correct it without additional training or data collection. Inspired by Wu et al. (2024), we measure the possibility of a prompt being hallucination-vulnerable by designing logic-contradictory prompts, which are easy to raise conflict with the original answer.

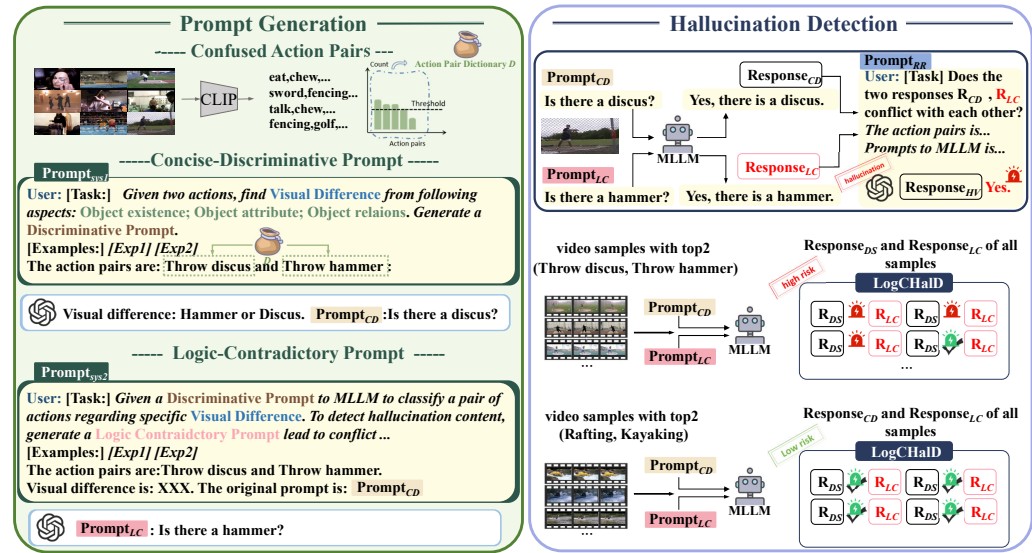

Figure 2: The overall framework of CDantiHalP. It has two key components: (1) concise-discriminative prompt, which activates MLLM recognition by highlighting distinguishing aspects of candidate categories rather than lengthy label lists, and (2) logic-contradictory hallucination detection (LogCHalD), which identifies potential hallucinations through contradiction-based prompting.

## 3 METHODOLOGY

In this paper, we address the challenge of zero-shot action recognition (ZSAR), which aims to identify the appropriate action label $c$ from a predefined set $\mathcal{C}$ for a given video $\mathcal{V}$, where the action labels in $\mathcal{C}$ are unseen by the prediction model during learning. To this end, we introduce a novel approach called **C**oncise, **D**iscriminative, and **anti-Hal**lucination **P**rompting (CDantiHalP), powered by LLMs (specifically GPT-3.5) to enhance the capabilities of MLLMs in distinguishing between easily confused action pairs while mitigating hallucination. The overall pipeline of CDantiHalP is illustrated in Figure 2. In the following sections, we provide a detailed explanation of CDantiHalP.

### 3.1 CONCISE, DISCRIMINATIVE PROMPTING FOR CONFUSED ACTION PAIRS

Our empirical results demonstrate that the generative capabilities of MLLM (i.e., VILA Lin et al. (2024)) can significantly outperform CLIP in the video action recognition task. However, as discussed in previous sections, directly feeding all action labels into MLLM to select one as the final prediction proves to be suboptimal. This is likely due to the model's diminished performance when processing long token sequences (i.e., lengthy prompts). To address this, instead of requiring MLLM to choose from a large set of potential labels, we leverage its strength in comparing specific concept pairs based on video frames. This approach is motivated by our observation that most misclassifications occur between the top-1 and top-2 predicted actions, which are often closely related (as indicated by the top-1 accuracy of 76.1% vs. top-2 accuracy of 88.7% on UCF101 using CLIP). We refer to these pairs of top-2 predictions as confused action pairs.

To capitalize on this observation, we first construct an action pair dictionary containing frequently confused action pairs, denoted as $\mathcal{D} = \{(c_i, c_j)\}_{i,j}$, where $(c_i, c_j)$ is a frequently confused action pair. In practice, we input the video samples into a simple CLIP-based text-enhanced action recognizer (specifically, TEAR Bosetti et al. (2024)), and record the top-2 recognized categories as potential confused pairs. Next, we count the occurrence frequency of these potential confused pairs and formally record pairs that exceed a specified threshold $\lambda$ (set to 5 in the experiment) as confused pairs, adding them to the dictionary $\mathcal{D}$.

After constructing the dictionary of confused action pairs $\mathcal{D}$, we shift the problem from selecting a single label among many to distinguishing between two specific actions. This allows us to replace a lengthy token sequence of potential action categories with a more concise and discriminative prompt. To achieve this, we design an LLM prompt ($\mathbf{Prompt}_{sys1}$, details in the left part of Figure 2) that

asks LLM to: *1.* highlight the **visual difference** between given confused action pair $(c_i, c_j)$; *2.* provide a corresponding question to distinguish a specific category within the pair (we denote it as **Prompt**$_{CD}$); and *3.* whether the confused action pair exhibits primarily static or dynamic attributes (formulated as [image]/[video] tag). The response of LLM is then formulated as **visual difference** + **Prompt**$_{CD}$ + [image]/[video]. Specifically, **Prompt**$_{CD}$ is used as the prompt to query the MLLM given a video to be classified, while the **visual difference** is used to optimize against MLLM hallucinations, which will be discussed in more detail in Sec 3.2. Additionally, the [image]/[video] tag guides the MLLM on whether to focus more on static or dynamic characteristics for video understanding. More technically, [image] tag signifies that the action pair involves strong static attributes, meaning the MLLM can effectively distinguish between the actions using a single video frame. Conversely, [video] tag indicates that the action pair involves strong dynamic attributes, requiring the MLLM to process the entire video (i.e., all sampled frames) to accurately recognize the action. For example (left part of Figure 2), when asked by an action pair *(Throw Hammer, Throw Discus)*, LLM response **visual difference** + **Prompt**$_{CD}$ + [image]/[video] as "*Hammer or Discus*" + "*Is there a discus*" + [image], in which the presence of "*Hammer or Discus*" serves as the **visual difference**, with *"Is there a discus"* as the question input to the MLLM to assist in classification. The [image] tag indicates that distinguishing this action pair only requires a single frame.

### 3.2 LOGIC-CONTRADICTORY HALLUCINATION DETECTION

Given that hallucinations can occur in MLLMs in various forms, an important question arises regarding the trustworthiness of the initial response (**Response**$_{CD}$). Specifically, we consider: ***Is the generated discriminative response influenced by hallucinations?*** To address this concern, we propose a logic-contradictory hallucination detection framework (LogCHalD).

LogCHalD aims at challenging the MLLM's understanding (**Response**$_{CD}$) upon **Prompt**$_{CD}$. Specifically, when faced with a pair of confused action pairs, we use **Prompt**$_{sys1}$ to obtain their **visual difference**, the key question for identifying between the two categories **Prompt**$_{CD}$, and the [image]/[video] tag to indicate whether identification relies more on a single frame or the entire video. After MLLM has responded to **Prompt**$_{CD}$, we aim to construct a logically opposing prompt, **Prompt**$_{LC}$. By observing whether the MLLM's responses to both **Prompt**$_{CD}$ and **Prompt**$_{LC}$ are logically consistent, we can determine whether MLLM hallucination is occurring. Using the example in the left part of Figure 2 to illustrate the generation of **Prompt**$_{LC}$, consider the action pair *(Throw Hammer, Throw Discus)*. Here, the LLM response includes "*Hammer or Discus*" + "*Is there a discus*", where "*Hammer or Discus*" represents a set of mutually exclusive concepts—meaning that likely to be confused, and implies that a video cannot contain both a hammer and a discus simultaneously. Therefore, we replace "discus" in "*Is there a discus*" with "hammer" to generate the logic-contradictory prompt (**Prompt**$_{LC}$), i.e., "*Is there a hammer*". For automatically conducting the generation of **Prompt**$_{LC}$, we introduce **Prompt**$_{sys2}$ to prompt LLM to conduct the aforementioned process.

Then, if the MLLM responds with the same answer to both **Prompt**$_{CD}$ and **Prompt**$_{LC}$, we conclude that a hallucination has occurred. Here, to further automate this process, we introduce a logic verification prompt (**Prompt**$_{LV}$) that prompts the LLM to systematically analyze potential hallucinations and generates the hallucination verification response (**Response**$_{HV}$ $\in$ *{Yes, No}*) to recognize the occupation of hallucination.

At this point, for a given confused action pair associated with a set of video samples, we can obtain the hallucination verification response (**Response**$_{HV}$) for each video or each video frame. Next, we calculate the proportion $r$ of hallucination verification responses where **Response**$_{HV}$=*Yes* for each confused action pair in the constructed dictionary $D$. Recall that the concise-discriminative prompt **Prompt**$_{CD}$ has highlighted the static (marked by [image]) or dynamic (marked by [video]) attribute for each action pair. As a result, for action pairs requiring the entire video input, the proportion $r$ is computed across all sampled videos, while for those requiring image frame input, $r$ is calculated from all individual frames. To clearly indicate the risk of hallucination of each action pair, we propose to categorize all the action pairs into two risk levels, i.e., low-risk group ($r \leq \theta$) and high-risk group ($r > \theta$). Here, the threshold $\theta$ is empirically set to 0.8. This categorization helps identify which action pairs are more prone to hallucinations under the current video/frame input and the generated discriminative prompts, thereby allowing us to focus on refining the classification results while minimizing the influence of hallucinations.

### 3.3 Hallucination Mitigation and Recognition Refinement

In this part, we will illustrate the refining pipeline of the top-2 action classification results by concise, discriminative prompting during inference time in detail. The entire process proceeds as shown in Algorithm 1. It is important to note that our proposed CDantiHalP approach can be applied to the predictions of any baseline model. The confused action pair dictionary $D$, $P$ containing $Prompt_{CD}$ for each pair in $D$, and the hallucination risk of each $Prompt_{CD}$ are all pre-gathered and fixed. Specifically, for a given video, we begin with obtaining the top-2 action candidates denoted as $c_1, c_2$. If $c_1, c_2$ exists in $D$, the corresponding $Prompt_{CD}$ for this action pair is recalled from $P$ (**Step 1**). We then evaluate the risk level of $Prompt_{CD}$ by comparing it with the given threshold $\theta$ and skipping those above it (**Step 2**). $Response_{CD}$ is acquired by feeding the MLLM with qualified $Prompt_{CD}$ and the original video $v$. Considering that $Prompt_{CD}$ can be either frame level or video level, we apply frame level $Prompt_{CD}$ to each sampled frames of $v$ resulting in a set of responses from the MLLM as $Response_{CD}$ while the $Response_{CD}$ of video level $Prompt_{CD}$ consists of only one response according to the complete video content (**Step 3**). Finally, we prompt LLM with $Prompt_{RR}$ to select the optimal action from $c_1, c_2$ based on the $Response_{CD}$. Overall, by leveraging the combined power of MLLM responses and LLM-based refinement, our CDantiHalP framework enhances the reliability of action recognition, especially in distinguishing between confused action pairs, while minimizing the influence of hallucinations.

---

**Algorithm 1** Refinement Process

---

**Require:** Pre-built confused action pair dictionary $D$, set of $Prompt_{CD}$ for each pair $P$, video $v$ and corresponding top-2 classification results $(c_1, c_2)$, **LLM, MLLM**
**Ensure:** Optimal action classification result
  1: **Step 1: Check if $c_1, c_2$ belongs to the confused action dictionary**
  2: **if** $(c_1, c_2) \in D$ **then**
  3:    Retrieve the corresponding $Prompt_{CD}$ from $P$
  4:    **Step 2: Evaluate the risk of $Prompt_{CD}$**
  5:    **if** $\text{Risk}(Prompt_{CD}) <$ threshold $\theta$ **then**
  6:      **Step 3: Query MLLM with $Prompt_{CD}$ and $v$**
  7:      $Response_{CD} = \text{MLLM}(v, Prompt_{CD})$
  8:      **Step 4: Select the best action**
  9:      $c = \text{LLM}(Prompt_{RR}, Response_{CD}, c_1, c_2)$
10:      Return $c$
11:    **else**
12:      Return $c_1$ (default action)
13:    **end if**
14: **else**
15:    Return $c_1$ (default action)
16: **end if**

---

## 4 Experiments

### 4.1 Datasets and Implementation Details

We conduct experiments on widely used ZSAR benchmarks UCF101 Soomro (2012), HMDB51 Kuehne et al. (2011), and Kinetics-600 (K600) Carreira et al. (2018). To ensure comparability, we adopt the same testing protocol as the previous works Lin et al. (2023); Rasheed et al. (2023). The following configuration is for the main results in our paper. We uniformly sample 16 frames from each video to align with the state-of-the-art methods and report the top-1 accuracy (%). We implement our method on multiple prevalent MLLMs, including VILA Lin et al. (2024), LLaVA-NeXT-Video Zhang et al. (2024) and VideoLLaMA2 Cheng et al. (2024a), all of which support both frame-level and video-level inputs. For fair comparison, we use the 7B version of all the MLLMs as each MLLM has a corresponding variant. GPT-3.5 is used as the LLM for generating $\mathbf{Prompt}_{CD}$, $\mathbf{Prompt}_{LC}$, $\mathbf{Prompt}_{LV}$ and $\mathbf{Prompt}_{RR}$.

### 4.2 Comparison with State-Of-The-Arts

We present our ZSAR results in Table 1, which are based on the pre-recognition outputs of the text-enhanced CLIP-based method TEAR Bosetti et al. (2024), and compare our method with many

existing SOTAs, including (1) uni-modal zero-shot video recognition methods, (2) CLIP adaption methods, (3) generative model-based methods, and (4) Training-free methods. First, our approach achieves the highest top-1 accuracies across all three benchmarks, significantly outperforming most training-based methods. Second, amongst these training-free methods, our post-refinement strategy yields substantial improvements. For instance, compared to the used pre-recognition baseline TEAR, our method shows top-1 accuracy gains of 4.7, 3.3 and 1.2 on HMDB51, UCF101, and K600, respectively. Additionally, in the supplementary material, we provide results using different pre-recognition baselines, demonstrating our method's strong cross-dataset and cross-baseline generalization capabilities.

Table 1: Comparison of zero-shot performance on various datasets.

| Method | Training | Foundations | Frames | HMDB51 | UCF101 | K600 |
|---|---|---|---|---|---|---|
| *Uni-modal zero-shot video recognition models* | | | | | | |
| GA | ✔ | C3D | — | 19.3 | 17.3 | — |
| ER-ZSAR Chen & Huang (2021) | ✔ | TSM | 16 | 35.3 | 51.8 | 42.1 |
| JigsawNet Qian et al. (2022) | ✔ | R(2+1)D | 16 | 38.7 | 56.0 | — |
| *Adapting pre-trained CLIP* | | | | | | |
| XCLIP Ni et al. (2022) | ✔ | ViT-B/16 | 32 | 44.6 | 72.0 | 65.2 |
| A5 Ju et al. (2022) | ✔ | ViT-B/16 | 32 | 44.3 | 69.3 | 55.8 |
| ActionCLIP Wang et al. (2023b) | ✔ | ViT-B/16 | 32 | 45.4 | 58.3 | 66.7 |
| DIST Qing et al. (2023) | ✔ | ViT-B/16 | 32 | 55.4 | 72.3 | — |
| Vita-CLIP Wasim et al. (2023) | ✔ | ViT-B/16 | 8/32 | 48.6 | 75.0 | 67.4 |
| ViFi-CLIP Rasheed et al. (2023) | ✔ | ViT-B/16 | 32 | 51.3 | 76.8 | 71.2 |
| MAXI Lin et al. (2023) | ✔ | ViT-B/16 | 16/32 | 52.3 | 78.2 | **71.5** |
| Text4Vis Wu et al. (2023a) | ✔ | ViT-L/14 | 16 | — | — | 68.9 |
| M$^2$-CLIP Wang et al. (2024b) | ✔ | ViT-B/16 | 32 | — | 78.7 | — |
| *Genreative models* | | | | | | |
| REST Bulat et al. (2022) | ✔ | BlipLi et al. (2022) | 16 / 32 | 49.7 | 69.1 | 29.6 |
| Regen Bulat et al. (2023) | ✔ | GITWang et al. (2022) | 16 / 32 | 55.1 | 76.4 | 38.2 |
| *Training-free methods* | | | | | | |
| BLIP2+Claude-1 Hanu et al. (2023) | ✘ | MLLM+LLM | — | — | 63.0 | — |
| TEAR Bosetti et al. (2024) | ✘ | ViT-B/16 | 16 | 50.8 | 76.2 | 70.3 |
| **Ours** (LLaVA-NeXT-Video) | ✘ | MLLM+LLM | 16 | 54.7 | **79.5** | 71.2 |
| **Ours** (VideoLLaMA2) | ✘ | MLLM+LLM | 16 | 53.8 | 78.7 | 71.2 |
| **Ours** (VILA) | ✘ | MLLM+LLM | 16 | **55.5** | 79.3 | **71.5** |

## 4.3 ABLATION STUDY

In this part, we perform ablations to validate our main designs, including the effect of concise-discriminative prompting, and the effect of hallucination mitigation. We use VILA Lin et al. (2024) as the MLLM in the rest of the experiments.

Here, we examine the effect of the proposed concise-discriminative prompt $\mathbf{Prompt}_{CD}$ by answering three important research questions (RQ):

Table 2: Performance comparison between our prompt and CLIP prompt.

| Prompt | Foundations | UCF101 | HMDB51 |
|---|---|---|---|
| Original | CLIP | 69.9 | 38.0 |
| Ours | CLIP | 74.2(+4.3) | 47.3(+9.3) |
| | MLLM | 79.3(+9.4) | 55.5(+17.5) |

Table 3: Performance comparison between different prompting strategies.

| Prompt | Visual Input | Act. Num. | UCF101 | HMDB51 |
|---|---|---|---|---|
| One-Many | Frame | 2 | 77.2 | 52.9 |
| | | 5 | 77.6 | 53.0 |
| | | All | 66.1 | 50.2 |
| | Video | 2 | 77.2 | 52.9 |
| | | 5 | 77.9 | 53.2 |
| | | All | 61.4 | 50.7 |
| Ours | Automatic | 2 | 79.3 | 55.5 |

Table 4: Performance comparison between prompting and captioning.

| Prompt | Visual Input | Act. Num. | UCF101 | HMDB51 |
|---|---|---|---|---|
| Captioning | Frame | 0 | 72.3 | 51.6 |
| | | 2 | 77.3 | 53.0 |
| | | 5 | 77.8 | 53.4 |
| | Video | 0 | 72.3 | 51.1 |
| | | 2 | 77.5 | 53.4 |
| | | 5 | 78.2 | 53.5 |
| Ours | Automatic | 2 | 79.3 | 55.5 |

Table 5: Performance improvement of our image and video level prompting.

| Prompt Type | Method | UCF101 | HMDB51 |
|---|---|---|---|
| Image | TEAR Bosetti et al. (2024) | 75.2 | 59.6 |
| | Ours | 86.6(+11.4) | 64.6(+5.0) |
| Video | TEAR Bosetti et al. (2024) | 65.7 | 68.5 |
| | Ours | 76.1(+10.4) | 79.4(+10.9) |

**RQ1. Does the generated concise-discriminative prompt clearly distinguish the confused actions?** To verify the distinguishability of $\mathbf{Prompt}_{CD}$, we test it with CLIP. For example, for the pair (*Parallel Bars, Uneven Bars*), $\mathbf{Prompt}CD$ generates: "Are the bars positioned at the same height?"

and its logical counterpart $\mathbf{Prompt}LC$: "Are the bars positioned at different heights?". These are reformulated into statements and encoded by CLIP; the video embedding is then compared to each text embedding via cosine similarity for prediction. As shown in Table 2, $\mathbf{Prompt}CD$ significantly outperforms the standard prompt "*a photo of action*" (+4.3 on UCF101, +9.3 on HMDB51). Moreover, integrating MLLM with $\mathbf{Prompt}_{CD}$ yields even larger gains (+9.4 and +17.5, respectively), confirming the superior discriminative power of our approach.

**RQ2. Is the proposed concise-discriminative prompting better than the prompting strategy of selecting one from many?** The core idea of our method is to replace the difficult task of choosing from many labels with the simpler task of distinguishing between action pairs, using concise-discriminative prompts that align with MLLMs' strengths. Table 3 shows two key findings: (1) narrowing the candidate pool (e.g., to 2 or 5 labels) significantly boosts performance by reducing prompt complexity and improving focus, and (2) our concise-discriminative prompting clearly outperforms the one-to-many strategy, highlighting its effectiveness.

**RQ3. Is the proposed concise-discriminative prompting better than the prompting strategy of captioning?** Another way to use MLLMs for ZSAR is as visual captioners, where the model generates video descriptions that are then classified by an LLM. In our setup, we vary the number of provided labels (0, 2, or 5). Table 4 shows two key findings: (1) omitting labels reduces accuracy, as the MLLM struggles to focus on action content; and (2) our concise-discriminative prompts with anti-hallucination strategies yield significant gains. This captioning approach parallels our framework, further validating the effectiveness of our prompt design. To further validate image- and video-level prompting, Table 5 reports accuracy on videos refined by concise-discriminative prompts. Compared to overall dataset results, these refinements yield larger gains, averaging +9.42 across datasets, clearly demonstrating the strength of our method.

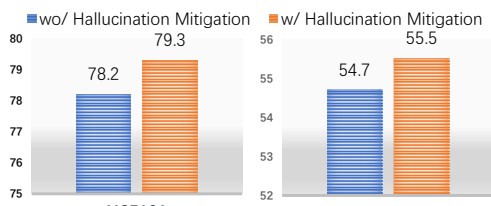

Figure 3: Performance change w/wo hallucination mitigation.

Table 6: Performance comparison of applying our methods on other baselines.

| Method | UCF101 | HMDB51 |
|---|---|---|
| ActionCLIP Wang et al. (2023b) | 73.7 | 50.0 |
| **+Ours** | 76.3(+2.6) | 51.6(+1.6) |
| XCLIP Ni et al. (2022) | 72 | 44.6 |
| **+Ours** | 74.9(+2.9) | 47.8(+3.2) |

To mitigate the impact of hallucination, we initially require the LLM to generate concise and discriminative prompts ($\mathbf{Prompt}_{CD}$) for clearly distinguishing between a confused action pair. In the refinement phase, we introduce a specific hallucination mitigation strategy based on the risk grouping results from LogCHalD. Specifically, we categorize the video inputs into either low-risk or high-risk groups based on their confused action pairs identified. For action pairs categorized as high-risk, we determine that the $\mathbf{Prompt}_{CD}$ and the current video qualities are insufficient to prevent hallucinated responses. Consequently, we refrain from using the MLLM's outputs for action recognition in these cases. For low-risk action pairs, when constructing the final LLM-based recognition refinement prompt $\mathbf{Prompt}_{RR}$, we additionally require it to account for the consistency among responses and follow the principle of "minority yielding to the majority". Figure 3 shows the performance changes w/wo the hallucination mitigation strategy applied in the refinement phase. The results demonstrate that implementing this hallucination mitigation significantly enhances overall performance, validating the effectiveness of our approach in improving action recognition accuracy while minimizing the impact of hallucination.

## 4.4 THE VERSATILITY OF OUR METHOD

Our method features training-independence and post-refinement capabilities, ensuring performance improvement on any baseline model. Here, we provide more results for another two baseline models, XCLIP and ActionCLIP, on UCF101 and HMDB51 datasets in Table 6. As observed, our refinement method achieves performance improvements ranging from 1.6 to 3.2 for the two baseline models, demonstrating its effectiveness and feasibility in enhancing action recognition.

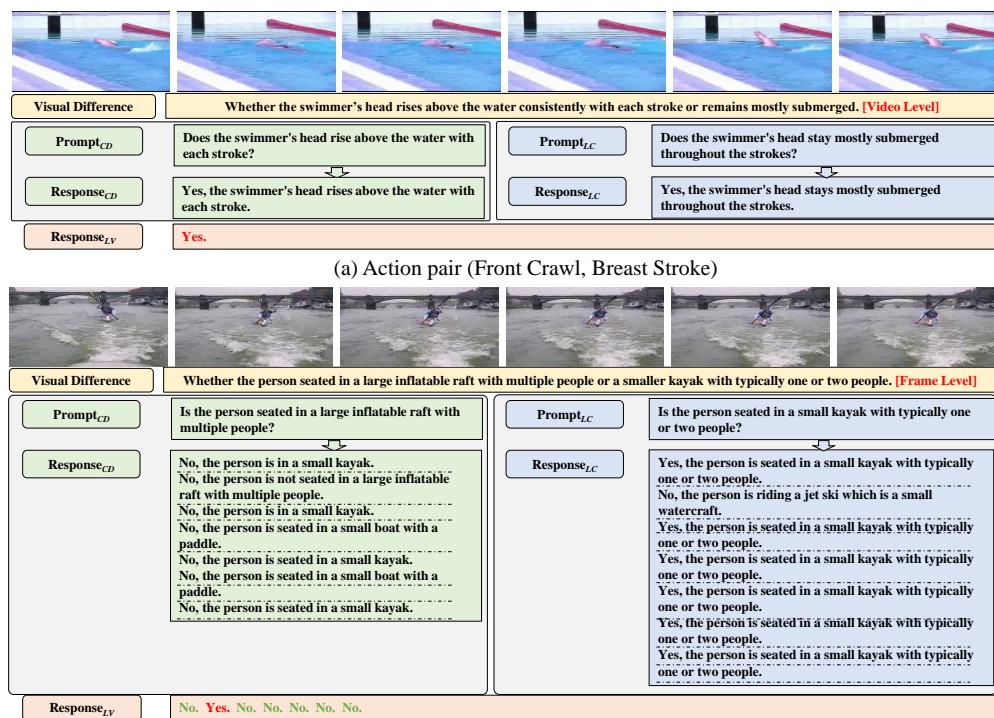

(a) Action pair (Front Crawl, Breast Stroke)

(b) Action pair (Rafting, Kayaking)

Figure 4: The processing results of different examples.

## 4.5 EXAMPLE DEMONSTRATION

In Figure 4, we show two cases to visually demonstrate the performance of CDantiHalP in generating concise-discriminative prompts and detecting hallucinations using logic-contradictory prompts. For the confused action pair "(*Front Crawl, Breast Stroke*)" (shown in Figure 4(a)), our method generates the concise-discriminative prompt $\mathbf{Prompt}_{CD}$ that highlights the key visual distinction, "*Whether the swimmer's head rises above the water consistently with each stroke or remains mostly submerged*", and accordingly formulates the question, "*Does the swimmer's head rise above the water with each stroke?*", prompting the MLLM to respond. To test for hallucinations, we automatically generate a logic-contradictory prompt $\mathbf{Prompt}_{LC}$, "*Does the swimmer's head stay mostly submerged throughout the strokes?*", to obtain the second response from MLLM. In this example, the MLLM produces the same response "*Yes*" to both contradictory questions, indicating the presence of hallucinations. Upon further analysis of the video content and prompts, we infer that the hallucinations may stem from the low quality of video frames. Moreover, the action of "(*Front Crawl, Breast Stroke*)" requires temporal context for understanding, thus is marked with "[Video]" to prompt the MLLM to consider the full video sequence. Figure 4(b) shows that most answers successfully pass the hallucination detection.

## 5 CONCLUSION

In this paper, we have introduced the regime of CDantiHalP, which aims to generate concise-discriminative prompts to enhance MLLM performance in distinguishing confused action pairs, while also addressing hallucination detection and mitigation to improve ZSAR. By shifting the focus from selecting among many labels to distinguishing between pairs, it enables MLLMs to interpret actions more effectively. The concise-discriminative prompts can explicitly focus on distinguishing two actions, reducing the probability of generating hallucinogenic answers. Hallucination detection is achieved through a logic-contradictory verification strategy, with risk levels assigned to confused pairs. Additionally, a training-free recognition refinement strategy uses the generated prompts and the hallucination mitigation instruction to ensure response consistency. In the experiment, we examine and demonstrate the effectiveness of the proposed concise-discriminative prompting and hallucination mitigation strategies in various ZSAR benchmarks.

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

# A  APPENDIX

In the supplementary material, we provide additional results analyzing the impact of hyper-parameters on constructing the confused action pair dictionary, along with the resulting action pairs. We also present detailed prompt designs with illustrative examples to demonstrate their role and functionality. Finally, we discuss the limitations of our method and outline directions for future work. For the use of LLMs, we only use them for language polishing.

## A.1  ABLATION STUDY ON HYPER-PARAMETERS

In building the confused action pair dictionary $\mathcal{D}$, we randomly sample a subset of test videos, count the frequency of top-2 co-occurring action pairs, and retain those exceeding a threshold $\lambda$. We evaluate the impact of both subset size and $\lambda$. (1) **Results with different video samples**: Figure 5(a) shows results on UCF101 and HMDB51 with 500, 1,000, and 1,500 samples. Accuracy improves with larger subsets (UCF101: 78.5%→79.4%; HMDB51: 55.3%→55.9%), as more samples yield richer, more representative action pairs. However, gains diminish as subset size grows. (2) **Results with different thresholds** $\lambda$: Figure 5(b) reports results for $\lambda = 3, 5, 10$. Accuracy remains stable at lower thresholds (UCF101: 79.3%, HMDB51: 55.5%) but drops when $\lambda = 10$ (77.2% and 55.0%), showing that overly strict thresholds exclude useful pairs and hurt performance.

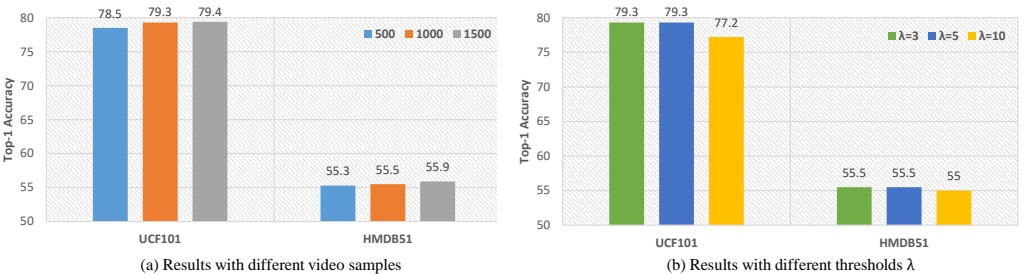

(a) Results with different video samples          (b) Results with different thresholds $\lambda$

Figure 5: Performance changes with different hyper-Parameters.

## A.2  PROMPT DETAILS AND MORE EXAMPLES

We present the details of prompts in our method and use examples to demonstrate their role and functionality. Figure 6 outlines the used LLM prompts, including $\mathbf{Prompt}_{sys1}$, $\mathbf{Prompt}_{sys2}$, $\mathbf{Prompt}_{LV}$ and $\mathbf{Prompt}_{RR}$. Particularly, "Visual difference" and $\mathbf{Prompt}_{CD}$ are generated by querying the LLM (GPT-3.5) using the system prompt $\mathbf{Prompt}_{sys1}$. The tag"[Video]/[Image]"

Table 7: The obtained action pairs with different numbers of sampled videos. The "shared" row indicates the overlapping action pairs of the three different sets.

| Num of Videos | Obtained Action Pairs |
| --- | --- |
| **HMDB51** | |
| 500 | *(shoot bow, sword exercise)* |
| 1,000 | *(hit, punch), (sit, stand), (golf, throw), (golf, turn), (draw sword, sword), (drink, smoke), (climb stairs, jump), (run, walk), (brush hair, clap), (pullup, pushup), (hit, shoot gun)* |
| 1,500 | *(climb stairs, walk), (shoot bow, sword exercise), (brush hair, talk), (golf, throw), (sit, stand), (drink, smoke), (hit, shoot gun), (run, walk), (draw sword, sword), (climb, jump), (climb, pullup), (kiss, talk), (ride bike, run), (punch, sword exercise), (brush hair, clap), (climb stairs, jump), (brush hair, laugh), (brush hair, smile), (brush hair, smoke), (drink, eat), (brush hair, wave), (swing baseball, sword exercise), (jump, run), (climb stairs, run), (brush hair, kiss)* |
| Shared | *(dribble, shoot ball), (kick, kick ball), (sword, sword exercise), (fencing, sword exercise), (shoot bow, shoot gun), (drink, pour), (jump, ride horse), (push, ride bike), (climb, climb stairs), (hit, sword), (shoot bow, sword), (dive, jump), (stand, walk)* |
| **UCF101** | |
| 500 | *(Swing, Trampoline Jumping), (Drumming, Playing Dhol)* |
| 1,000 | *(Field Hockey Penalty, Soccer Penalty), (Brushing Teeth, Shaving Beard), (Jumping Jack, Lunges), (High Jump, Pole Vault), (Apply Lipstick, Brushing Teeth), (Drumming, Playing Daf), (Biking, Skate Boarding), (Horse Riding, Pommel Horse), (Bench Press, Body Weight Squats), (Typing, Writing On Board), (Boxing Punching Bag, Punch), (Fencing, Hammer Throw), (Baby Crawling, Push Ups), (Playing Flute, Playing Piano), (Rope Climbing, Uneven Bars), (Golf Swing, Hammer Throw)* |
| 1,500 | *(Cricket Bowling, Cricket Shot), (Field Hockey Penalty, Soccer Penalty), (Drumming, Playing Daf), (High Jump, Pole Vault), (Brushing Teeth, Shaving Beard), (Boxing Punching Bag, Punch), (Horse Riding, Pommel Horse), (Typing, Writing On Board), (Apply Lipstick, Brushing Teeth), (Skiing, Skijet), (Cliff Diving, Skijet), (Golf Swing, Hammer Throw), (Jumping Jack, Lunges), (Biking, Skate Boarding), (Baby Crawling, Push Ups), (Clean And Jerk, Lunges), (Hammering, Knitting), (Fencing, Hammer Throw), (Playing Flute, Playing Piano), (Hammer Throw, Javelin Throw), (Drumming, Playing Dhol), (Fencing, Pommel Horse), (Bench Press, Body Weight Squats), (Playing Guitar, Playing Piano), (Skate Boarding, Walking With Dog), (Skate Boarding, Skijet), (Rope Climbing, Uneven Bars), (Bench Press, Pull Ups), (Biking, Lunges), (Hammering, Writing On Board), (Blowing Candles, Pizza Tossing), (Jumping Jack, Walking With Dog), (Rafting, Skijet)* |
| Shared | *(Playing Dhol, Playing Tabla), (Apply Eye Makeup, Apply Lipstick), (Skijet, Surfing), (Playing Guitar, Playing Sitar), (Playing Cello, Playing Violin), (Ice Dancing, Salsa Spin), (Horse Race, Horse Riding), (Blow Dry Hair, Haircut), (Hand Stand Pushups, Handstand Walking), (Billiards, Table Tennis Shot), (Bench Press, Clean And Jerk), (Playing Flute, Playing Violin), (Kayaking, Rafting), (Billiards, Bowling), (Body Weight Squats, Lunges), (Skijet, Sky Diving), (Hand Stand Pushups, Push Ups), (Field Hockey Penalty, Frisbee Catch)* |

indicates the input format of MLLM, which is also automatically highlighted. $\mathbf{Prompt}_{LC}$ serves as the logic contradictory trigger for the MLLM to respond based on video content, derived from prompting the LLM with $\mathbf{Prompt}_{sys2}$. Consequently, $\mathbf{Response}_{CD}$ and $\mathbf{Response}_{LC}$ represent the corresponding outputs from the MLLM. Using the LLM-based prompt $\mathbf{Prompt}_{LV}$, we verify the logical conflict between the two responses, resulting in the hallucination verification output $\mathbf{Response}_{LV}$ (yes or no). Importantly, all of the above prompting operations are performed in a single processing step to assess the risk level of each confused action pair. Finally, during the refinement of the baseline recognition results, for a specific video whose top-2 actions are in the confused action pair dictionary $D$, we only need to: (1) prompt MLLM with $\mathbf{Prompt}_{CD}$ to obtain $\mathbf{Response}_{CD}$, and (2) use the recognition refinement prompt $\mathbf{Prompt}_{RR}$ to trigger the LLM based on the $\mathbf{Prompt}_{CD}$, "Visual Difference" and $\mathbf{Response}_{CD}$ for action prediction.

Figures 7–8 illustrate how CDantiHalP generates concise-discriminative prompts and detects hallucinations via logic-contradictory prompts. For the confused pair "(*Uneven Bars, Parallel Bars*)" (Fig. 7), our method creates $\mathbf{Prompt}CD$: "*Are the bars positioned at the same height?*" and its contradictory $\mathbf{Prompt}LC$: "*Are the bars positioned at different heights?*". The MLLM responds "*No, ...*" to both, revealing a hallucination likely caused by complex visual elements in the video. Moreover, $\mathbf{Prompt}_{CD}$ adaptively identifies whether a pair requires static (marked as "[Image]") or dynamic (marked as "[Video]") cues. For example, "(*Hand Stand Pushups, Handstand Walking*)" (Figure 8) depends on temporal context and is thus marked "[Video]" to ensure sequence-level reasoning.

---

**Prompt_sys1**

My video classification model fails to accurately classify some pairs of video categories for example "apply lipstick" and "apply eye makeup".
I need you to generate some questions to ask a multi-modal large language model and the answer should be effective enough to classify the video category pair. Meanwhile, tell the MLLM need to answer the question through the whole video or one image in the video.
I need you to answer with format:  the visual difference you find + [image]/[video] + question
You can design a question from the following aspect or based on the following ideas:
1. Object existence, if there are some objects which indicate the action category
2. Visual attributes, like human body posture, the visual appearance of a human, or the status of human joints
3. Object relationship, including interaction patterns between different objects or human bodies like "the cup is held in the hand", "arm is above head"
For example, to distinguish the action pair: "apply lipstick" and "apply eye makeup" your reply can be:
"whether the tool the person holds is touching the person's lip or eye" [image] "Is the tool the person holding touching the person's lip?"
For example, to distinguish the action pair: "throwing discus" and "throwing hammer" your reply can be:
"whether there is a hammer or a discus" [image] "Is there a discus?"
Due to poor video quality and blurriness. Please prioritize selecting larger targets or motions that are more prominent and easier to discern clearly.  Please make this question as clear, simple, and concise as possible. Please make sure your question can apply to general and a wide range of situations
Now the two actions are [action1],[action2]:

---

**Prompt_sys2**

To select the most likely action from two action candidates in the video, I ask a question to a multi-modal large language model and use its response to the question on video frames to classify. The question is designed considering the visual difference of the actions.
However the multi-modal large language model faces the hallucination problem, they tend to give positive answers or describe content that does not exist in the image. I need you to generate a hallucination detection question. If the multi-modal large language model outputs a positive answer to this question, it would raise a conflict with the original positive answer.Give me the question only without any more words.
For example, the original question is  "Is there a discus?", the original positive answer is "Yes, there is" which proves the right action is "throwing discus". To raise conflict with this, you can assume the real action is "throwing hammer" and there is a hammer in the image then raise questions about the hammer in the image. Your answer can be: "Is there a hammer?" "Describe the hammer" or "In which hand does the person hold the hammer?". The positive answer to these questions proves the existence of the hammer and suggests that the right action is "throwing hammer" which is contradictory to the original positive response.
For example: The actions to be distinguished are "apply lipstick" and "apply eye makeup". The visual difference is "whether the tool the person holds is touching the person's lip or eye" The original question is "Is the tool the person holding touching the person's lip?"
Your answer can be: "Is the tool the person holding touching the person's eye?"
Now The actions to be distinguished are [action1, action2]. The visual difference is [visual difference]. The original question is [question]:

---

**Prompt_LV**

To select the most likely action from two action candidates in the video, I ask multi-modal large language model two questions. These questions are proposed based on the key visual difference between these two actions. I will give you the responses of the multi-modal large language model to these questions with the same input. You should infer these questions are contradictory for example they stand for different actions and then answer yes or no.
For example, the actions to be distinguished are "throwing discus" and "throwing hammer", their visual difference is "whether there is a hammer or a discus". The two questions are 1."Is there a discus?" and 2."Is there a hammer?" The two responses are: 1. Yes, there is a discus 2. Yes, there is a hammer. These two responses suggest different actions "throwing discus" and "throwing hammer" in the video respectively which incur conflict. So you answer "yes".
Now, the actions to be distinguished are [action1], [action2], their visual difference is [visual difference].
The two questions are 1. [Prompt_CD] and 2. [Prompt_LC]
The two responses to these questions respectively are: 1. [Response_DS] 2. [Response_LC]
Are these two responses contradictory to each other?

---

**Prompt_RR**

To select the most likely action from two action candidates [action1], [action2] in the video. I ask a question [Prompt_CD], due to their visual difference [visual difference] to a multi-modal large language model. Here are the frame-level or video-level responses of the model:
[Response_DS]...
I need you to select the most likely human action in the video according to the responses.
Note 1: Video frames are randomly sampled from the entire video, so not all frames may come from key moments. Some frames might only depict the background and lack useful information for classification. Note 2: "If there are conflicting results between different frames, it may be due to the quality of a few frames, so please select the answer based on the principle of minority yielding to the majority."
You must select an action inside the two actions [action1, action2] provided and output an identical name of it.

---

Figure 6: The constructed prompts for LLM.

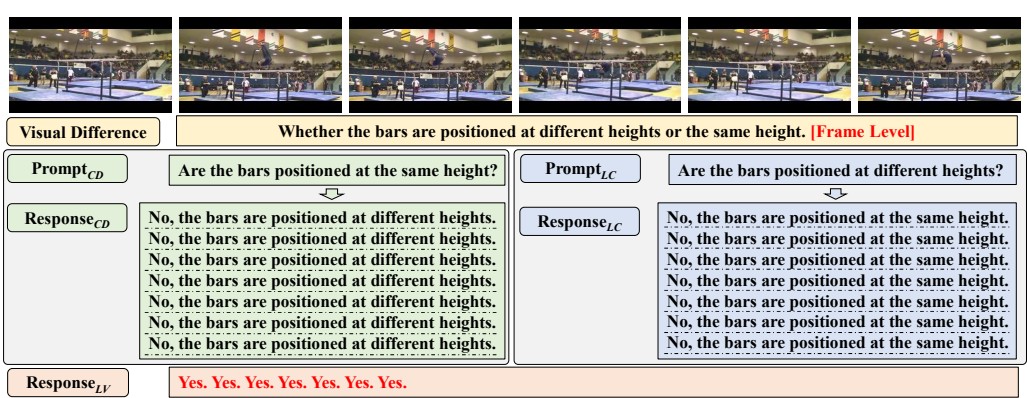

Figure 7: Example of action pair (Parallel Bars, Uneven Bars)

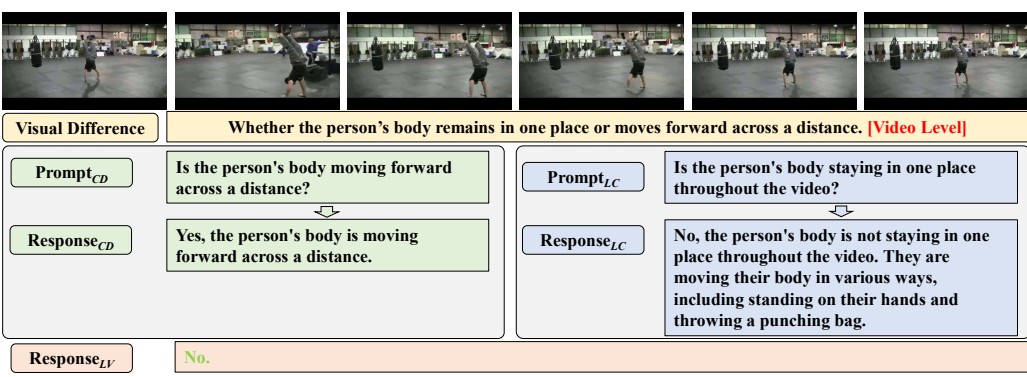

Figure 8: Example of action pair (Hand Stand Pushups, Handstand Walking)

### A.3 LIMITATION AND FUTURE WORK

Our CDantiHalP is currently limited to refining the top-2 predictions of a ZSAR baseline. The concise-discriminative prompt generation process focuses solely on differentiating two actions, which restricts its application to more actions. In future work, we aim to extend this approach to differentiate among a larger set of actions, such as top-3 or top-5 candidates. Another limitation is that for high-risk hallucination pairs, our current method does not provide strategies for improvement. Moving forward, we plan to address this by (1) prompting LLM to interpret the causes of high-risk hallucinations and (2) automatically revising the concise-discriminative prompts based on these interpretations.

