# OpenReview forum: "Leveraging MLLMs for Zero-Shot Action Recognition: Concise, Discriminative and anti-Hallucination Prompting"
_ICLR.cc/2026/Conference — Submitted to ICLR 2026_

### Official Review · Reviewer_sE3M · 2025-10-26

**Soundness:** 3
**Presentation:** 3
**Contribution:** 3
**Rating:** 6
**Confidence:** 3

**Summary:**

This paper addresses the problem of zero-shot action recognition (ZSAR) using multi-modal large language models (MLLMs) without additional training data. The authors identify two key limitations of current MLLM-based ZSAR methods:
1) Ineffective prompt design: lengthy or vague prompts that reduce discriminative power.
2) Hallucination:  incorrect or inconsistent predictions due to over-reliance on priors or noisy video frames.

the paper proposes CDantiHalP which is a training-free, post-refinement framework composed of:
1) Concise, Discriminative Prompting (PromptCD): Automatically generated prompts contrast confused action pairs (e.g., throw discus vs. throw hammer) rather than having the model choose from an entire label list.Prompts are tagged as [image]/[video] depending on whether static or dynamic cues are most relevant.
2) Logic-Contradictory Hallucination Detection (LogCHalD): a logic-contradictory prompt (PromptLC) to test the consistency of MLLM responses. If the model gives the same answer to contradictory queries, hallucination is detected.
3) Recognition Refinement: LLM reasoning (PromptRR) to reconcile responses and refine final predictions, prioritizing consistent, low-risk outputs.

Empirical results on UCF101, HMDB51, and Kinetics-600 show that CDantiHalP achieves state-of-the-art zero-shot performance and improves existing baselines (e.g., TEAR, XCLIP, ActionCLIP).

**Strengths:**

1) The paper includes comprehensive experiments and ablations validate every component (prompt design, hallucination mitigation, frame/video tagging, hyperparameters). The paper also evaluates multiple MLLM backbones (LLaVA-NeXT-Video, VideoLLaMA2, VILA) and demonstrates cross-model robustness.
2) CDantiHalP offers a practical, scalable enhancement for ZSAR pipelines without additional training cost. The logic-contradictory hallucination detection (LogCHalD) is a creative, lightweight mechanism for assessing hallucination risk without retraining.

**Weaknesses:**

1) The approach relies on a co-occurrence-based “confused pair dictionary” generated from baseline model outputs. This might limit applicability to new datasets or unseen classes where no prior co-occurrence statistics exist. A discussion of adaptive or online pair construction would be valuable.
2) For LogCHalD, quantitative metrics for hallucination detection accuracy are not explicitly reported. A confusion matrix or correlation between hallucination risk and performance degradation would make the claim stronger.
3) Since multiple MLLM queries are made per video (PromptCD and PromptLC, sometimes per frame), runtime cost could be substantial. The paper should include timing or computational analysis.

**Questions:**

1) How sensitive is CDantiHalP to the specific LLM used (e.g., GPT-3.5 vs. open-source LLMs)? Can smaller models (e.g., Vicuna, Mistral) reproduce the same quality prompts?
2) Could the dictionary be built dynamically during inference, e.g., via nearest-neighbor semantic similarity rather than fixed co-occurrence?  How does performance degrade if the dictionary is incomplete or noisy?
3) Have the authors evaluated LogCHalD’s ability to correctly identify hallucinations (precision/recall)? Does the threshold  θ=0.8 generalize across datasets, or was it tuned per dataset?

---

> ### Author Response · Authors · 2025-11-23
> **response to reviewer 3**
>
> We are grateful for your thorough review and constructive feedback. We will address all comments systematically with new experiments added as suggested. We hope the responses satisfactorily resolve your concerns.
>
> ## **Weakness 1** and **Question 2**
>
> - A discussion of **adaptive or online pair construction** would be valuable....
>
> - **Could the dictionary be built dynamically during inference**, e.g., via nearest-neighbor semantic similarity rather than fixed co-occurrence? ...
>
> ### **Response:**
>
> Thanks for pointing out this limitation. We propose an **Online Version** and obtain an accuracy of **79.0% on the UCF dataset** (a marginal 0.3% degradation compared with the original 79.3%). Descriptions are as follows:
>
> We first randomly shuffle the test dataset $V$ and sequentially process videos $v_i$, where $i = 1, 2, 3, \dots, n$ and $n = |V|$. We use the first 1000 videos as a subset $\hat{V}$ from $V$ for dynamic dictionary construction. During this process, we maintain a continuously updated **Confused Pair Dictionary** ($D$), a **Confusing Pair Pool**($P$), and a dynamically updated **Confused Pair Frequency Counter** ($count_{c_i,c_j}$), **Confused Pair Risk** for each PromptCD. $\lambda$ is the frequency threshold as in the main paper.
>
> For each video $v_i \in \hat{V}$:
>  1.  Obtain the top-2 prediction results $(c_1, c_2)$ for $v_i$.
>  2.  Increment the frequency counter: $count_{c_1,c_2} = count_{c_1,c_2} + 1$ or initiate the $count_{c_1,c_2}$ to 1.
>  3. **If** $(c_1, c_2)$ not in $P$: add $(c_1, c_2)$ to $P$
>  4. **If** $(c_1, c_2)$ does not exist in $D$:
>      *   **If** $count_{c_1,c_2} = \lambda$:
>          *   Generate $Prompt_{CD}$ and $Prompt_{LC}$ for $(c_1, c_2)$.
>          *   Initialize $risk_{c_1,c_2}$ to 0.
>          *   Add $(c_1, c_2)$ to $D$ and proceed to step 5.
>  5.  **If** $(c_1, c_2)$ exists in $D$:
>      *   Retrieve the corresponding $Prompt_{CD}$ and $Prompt_{LC}$.
>      *   Refine the top-2 prediction for $v_i$ and calculate the contradiction ratio.
>      *   Update the risk score: $risk_{c_1,c_2} = ((count_{c_1,c_2}-\lambda) * risk_{c_1,c_2} + ratio) / (count_{c_1,c_2} - \lambda + 1)$.
>
>
> After iterating through videos, we will have established the D and the associated **Pair Risk** for each PromptCD. We adopt the same strategy to refine the remaining videos as in the main paper.
>
> In our view, building D through nearest-neighbor semantic similarity represents a pre-caching strategy for constructing the confusion dictionary, which allows us to obtain D from only labels.
>
>
> ## **Weakness 2** and **Question 3**
>
> 1. **quantitative metrics for hallucination detection accuracy** are not explicitly reported...
> **evaluated LogCHalD’s ability** to correctly identify hallucinations (precision/recall)?
>
> 2. **Does the threshold θ=0.8 generalize** across datasets, or **was it tuned per dataset**?
>
> ### **Response:**
>
> We appreciate the reviewer for highlighting the missing evaluation and pointing out a valuable perspective. We supplement the precision evaluation of the hallucination detection method, performance under different risk threshold θ, and a dataset-adaptive threshold θ selection strategy.
>
> **Part1:** **Precision evaluation of hallucination detection**
> We **define hallucination as an inconsistency between the response of PromptCD and actual video content**. Detection is successful when PromptCD and PromptLC responses conflict logically. On UCF101 and HMDB51, we **achieved a hallucination detection precision of 85.94% and 81.25% respectively**. Since, by definition, any conflict between the two verification prompts signals an unreliable prediction (i.e., hallucination), there is no scenario of a false positive (mistakenly flagged hallucination). Consequently, we can only compute the precision metric.
>
>
> **Part2:** **Performance under different θ**
> Performance under different thresholds θ for UCF101 and HMDB51 datasets:
> | Threshold | 0.8  | 0.7  | 0.6  | 0.4  | 0.2  |
> |-----------|------|------|------|------|------|
> | UCF101    | 79.3 | 79.3 | 79.1 | 78.7 | 78.2 |
> | HMDB      | 55.5 | 55.5 | 55.3 | 54.9 | 54.7 |
>
> **Part3:** **Dataset-adaptive threshold θ selection strategy**
> We provide a simple and effective **dataset-adaptive threshold θ selection strategy**, which shows no degradation. We first **rank all PromptCDs in D** based on their risk of hallucination, then **retain the top 30%** with the lowest risk. This results in **thresholds of θ=0.65 for UCF101 and θ=0.71 for HMDB51**, while **maintaining performance without degradation (79.3% for UCF101 and 55.5% for HMDB51)**.

---

> ### Author Response · Authors · 2025-11-23
> **response to reviewer 3**
>
> ## **Question 1**
>
> - How sensitive is CDantiHalP to the specific LLM used (e.g., GPT-3.5 vs. **open-source LLMs**)? **Can smaller models (e.g., Vicuna, Mistral) reproduce the same quality prompts?**
>
> ### **Response:**
> Thank you for this insightful advice. **We have replaced GPT-3.5 with the smaller open-source model Qwen3-30B** (Qwen3-30B-A3B-Instruct-2507) for prompt generation and verification. On UCF101, it **achieves an accuracy of 78.9% ( GPT-3.5 79.3%, Baseline Tear 76.2%)**. **Analysis and observations** are as follows:
> We note that in the generation of PromptCD (Figure 6), the LLM is explicitly constrained to produce deterministic PromptCD based on three predefined aspects:
> - The existence of specific objects
> - Visual attributes, such as body posture or human appearance
> - Interaction relationships, e.g., "person holds a cup"
>
> This design (1) restricts the LLM's behaviour and exploration direction; (2) ensures that similar PromptCDs can be obtained across different language models, thereby enhancing the stability and generalizability of our overall approach; (3) these also serve as effective guidance for smaller-scale models, enabling them to effectively generate discriminative PromptCDs.
> Here we provide two examples listing the PromptCDs generated by GPT-3.5 and Qwen3-30B. These two models generate similar PromptCDs, concentrating on visual attributes (same height) and the existence of a specific object (beam).
>
> Pair: "Uneven Bars", "Parallel Bars"**
> GPT-3.5: "Are the bars positioned at the same height?"
> Qwen3-30B: "Are the two bars at the same height?"
>
> Pair: 'Uneven Bars', 'Balance Beam'**
> GPT-3.5: "Is the person performing on a balance beam?"
> Qwen3-30B: "Is there a single, narrow beam?"
>
>
> ##  **Question 2**
>
> - How does performance degrade **if the dictionary is incomplete or noisy**?
>
> ### **Response**
> Thanks for the insightful question regarding the dictionary's flexibility and robustness. Our analysis of the impact of an incomplete or noisy dictionary D is as follows:
>
> 1.If the **dictionary D is incomplete**, a video's **top-2 prediction pair may not be found in D**. Then that **sample**'s prediction **will simply not be refined**, nor will it incur any additional computational cost. When building dictionary D, we select action pairs with higher frequency to avoid such a situation.
>
> 2.If the **dictionary is noisy** (e.g., some unobserved pairs occur in the dictionary), **these noisy pairs will have virtually no effect on performance**. This is because **they will not be retrieved from D** to refine any actual top-2 prediction results, thus consuming no computational resources.

---

### Official Review · Reviewer_CBdX · 2025-10-30

**Soundness:** 2
**Presentation:** 2
**Contribution:** 2
**Rating:** 4
**Confidence:** 3

**Summary:**

The proposed method is a training-free, post-refinement pipeline for Zero-Shot Action Recognition that converts a multi-class problem into pairwise discrimination between likely-confused classes. It uses an LLM to auto-generate a concise, discriminative prompt (PromptCD) and a logic-contradictory prompt (PromptLC) to probe hallucinations (LogCHalD). At inference, it refines a baseline model’s top-2 using these prompts and a consistency rule.

**Strengths:**

1. Instead of  long, label-list prompts (token-heavy) it transforms to binary, feature-targeted questions which reduces prompt length/noise.
2.  The method  integrates with multiple MLLMs  and yields consistent gains—useful in low-resource or deployment-constrained settings.
3. Exhaustive ablation study.

**Weaknesses:**

1.  The confused-pair dictionary D is built by “randomly sampling a subset of test videos,” collecting top-2 co-occurrences, and keeping pairs above a certain λ. Ideally it would good to derive D from train/auxiliary data or a held-out validation set and freeze it before testing. It appears that the current settings is a test-aware meta-tuning of the prompt inventory.
2.  It seems that LogCHalD will work when objects are mutually exclusive within a clip, but can fail when both objects can co-occur (multi-person scenes, equipment in the background).
3. An adaptive or learned threshold (or cost-sensitive policy) would be preferable instead of the static threshold, especially given class imbalance and varying yes-bias across models. Since Different MLLMs may have different bias profiles., Also r’s distribution will shift across datasets.
4. The method requires multiple MLLM passes.

**Questions:**

Can you think of an easy solution for the adaptive threshold?  Also did you purposely exclude Something Something Dataset considering its long temporal dependence ?

---

> ### Author Response · Authors · 2025-11-23
> **response to reviewer 2**
>
> We appreciate your thoughtful comments. We will address each question and weakness, including additional experiments where recommended. We hope these responses can clarify our methodology and resolve your concerns.
>
> ## **weakness 1**
>
> - Ideally it would good to derive D from train/auxiliary data or a held-out validation set and freeze it before testing. **It appears that the current settings is a test-aware meta-tuning of the prompt inventory.**
>
> ### **Response:**
> Thank you for this valuable comment. Our method does **utilize test data** in collecting D **in an unsupervised manner**, requiring no label information. **The use of test data for certain adaptations is established** in the field of zero-shot action recognition. **As noted in [1]**, the authors propose to "re-position the semantic prototypes of unseen actions by matching them to **the distribution of all test videos**." We will emphasize this characteristic of our work in the revised paper.
>
> [1] Universal prototype transport for zero-shot action recognition and localization. Mettes, Pascal. IJCV 2023
>
> ## **weakness 2**
>
> - It seems that **LogCHalD** will work when objects are mutually exclusive within a clip, but can **fail when both objects can co-occur** (multi-person scenes, equipment in the background).
>
> ### **Response:**
> Thank you for raising this insightful observation. We want to note that **our hallucination detection method** exhibits a degree of effectiveness in **addressing these cases** and preventing possible harmful impact. Our analysis is as follows:
>
> (1)Using action pair (throw hammer, throw discus) as an example. The PromptCD is "Is there a discus in the image?". The logic-contradictory prompt PromptLC is "Is there a hammer in the image?". **Once both hammer and discus appear in a video $v_i$, both PromptCD and PromptLC will receive "yes" responses**. This contradiction **signals hallucination**. Accordingly, we skip refining $v_i$, thus preventing harmful impacts.
>
> (2)**We categorize this phenomenon as a form of generalized hallucination**, in which case PromptCD fails to accurately determine the actual action represented in a video.
>
> (3)**If this phenomenon is widespread in the dataset**, it indicates that **the PromptCD** is not proper for the dataset. This will be reflected in the risk level. If a PromptCD frequently triggers hallucination signals, it **will be discarded**. No additional analysis will be conducted for videos with this action pair and PromptCD, thereby avoiding all potential negative impacts—including computational resource consumption and performance degradation.
>
> (4)Finally, **we note that object existence is one of the three directions in generating PromptCD** (along with visual attributes and interaction relationships). Meanwhile, both the UCF101 and HMDB51 datasets are characterized by clear primary actions and clean backgrounds. Overall, our method achieves improvements of 4.7 and 3.3 on HMDB51 and UCF101, respectively, which demonstrates the applicability of our CDantiHalP.
>
> In our current design, we address the issue you raised by identifying and skipping such videos. We aim to develop more adaptive approaches for these cases in the future.

---

> ### Author Response · Authors · 2025-11-23
> **response to reviewer 2**
>
> ## **weakness 4**
> - The method requires **multiple MLLM passes**.
>
> ### **Response:**
>
> In the refinement stage, **each video requires only one turn of MLLM inference**. As **the hallucination risk of PromptCD is pre-determined**, no additional hallucination detection is needed. Only 20.24% of test videos undergo refinement after filtering high-risk PromptCDs, completing in 28 minutes overall.
>
> ## **Question 1**
> -  Also did you purposely exclude **Something Something Dataset**
>
> ### **Response:**
>
> Thank you for your reminder. Some of **our baseline models and comparative methods [1-4] were only evaluated on HMDB51, UCF101, and Kinetics-600 datasets** without including SSV2 analysis, which is also a **common configuration in zero-shot action recognition**. We have followed the same evaluation protocol. **We have conducted an experiment on the SSV2 dataset**. Since CLIP performs poorly on the SSV2 dataset (with only 0.15% top-1 and 0.94% top-5 accuracy), we developed CDantiHalP based on CAST [5]. Applied to CAST, our method **improves accuracy from 68.46% (baseline) to 68.93%**, demonstrating the generalizability of our approach. However, the improvement on SSV2 is more limited compared to the gains on UCF101 and HMDB51. This is mainly because recognizing action pairs in SSV2 requires capturing more dynamic visual cues. For instance, distinguishing between "Something falling like a rock" and "Dropping something onto something" depends on detecting whether an object falls heavily - a subtle distinction that poses challenges for current MLLMs.
>
>
> [1]Bulat, Adrian, et al. "ReGen: A good Generative zero-shot video classifier should be Rewarded." CVPR, 2023.
> [2]Bosetti, Massimo, et al. "Text-Enhanced Zero-Shot Action Recognition: A Training-Free Approach." ICPR, 2024.
> [3]Wang, Mengmeng, et al. "Actionclip: Adapting language-image pretrained models for video action recognition." TNNLS 2023.
> [4]Ni, Bolin, et al. "Expanding language-image pretrained models for general video recognition." ECCV, 2022.
> [5] Lee, Dongho, et al. "CAST: cross-attention in space and time for video action recognition." NIPS 2023

---

### Official Review · Reviewer_bpxS · 2025-11-01

**Soundness:** 3
**Presentation:** 3
**Contribution:** 3
**Rating:** 6
**Confidence:** 3

**Summary:**

This paper introduces CDantiHalP, a training-free, post-refinement framework designed to improve zero-shot action recognition (ZSAR) using multimodal large language models (MLLMs). The core idea is to replace vague, lengthy prompts with concise and discriminative ones (PromptCD) that focus on distinguishing confused action pairs. Furthermore, a logic-contradictory hallucination detection module (LogCHalD) is proposed to verify the consistency of responses and identify hallucinated outputs. Experiments conducted on UCF101, HMDB51, and Kinetics-600 using VILA, LLaVA-NeXT-Video, and VideoLLaMA2 demonstrate consistent performance gains across datasets. The method achieves state-of-the-art accuracy among training-free approaches and shows strong generalization across models and baselines.

**Strengths:**

1. The proposed Concise-Discriminative Prompting effectively exploits the comparative reasoning strength of MLLMs, turning a complex multi-label selection problem into pairwise action discrimination.

2. The Logic-Contradictory Hallucination Detection (LogCHalD) is a clever design that uses contradictory prompts to test the internal logical consistency of model outputs, providing an interpretable way to identify hallucinations.

3. The framework requires no fine-tuning or additional data, and can refine predictions from any existing ZSAR baseline, showing strong portability.

4.  The paper presents solid empirical evaluations with ablations, qualitative visualizations, and cross-model generalization studies.

**Weaknesses:**

1. While the paper reports impressive results on VILA (Lin et al., 2024), LLaVA-NeXT-Video (Zhang et al., 2024), and VideoLLaMA2 (Cheng et al., 2024), these models may no longer be at the frontier of video understanding. It would be valuable to test newer and stronger MLLMs, such as Qwen3-VL or InternVL2/3, which exhibit fewer hallucinations and stronger temporal modeling. Evaluating CDantiHalP on these modern models would better validate its robustness and contributions to the field.

2.  The current pipeline heavily relies on GPT-3.5 to generate Concise-Discriminative (PromptCD) and Logic-Contradictory (PromptLC) prompts, as well as to verify hallucination conflicts.
   This dependency increases inference-time latency and cost.
   The paper could discuss whether these components can be generated or approximated by smaller, open-source LLMs (e.g., 72B or 34B in scale) to reduce overhead without degrading performance.
   Besides, the verify step may potentially be done by rule-based methods without relying on large models since it involves checking for logical contradictions. Discussing such alternatives would enhance practicality.

3.  Although the method is “training-free,” the number of GPT and MLLM calls per sample may be high. Quantitative analysis of latency, API cost, and runtime complexity may be missing.

4.  The experiments are limited to three standard action recognition datasets. Evaluating on real-world long video datasets (e.g., ActivityNet-QA, Something-Something V2) or multimodal reasoning benchmarks may further establish generality.

**Questions:**

1. Add experiments on modern and stronger models to strengthen the claim of general applicability.

2. Explore cost-reduction strategies using smaller open-source models for generating and verifying prompts, or precomputing a reusable prompt library to reduce the GPT query burden.

3. Provide detailed efficiency analysis by reporting average computation time and API costs for prompt generation and hallucination detection, comparing them against simpler baselines.

4. Extend evaluation beyond ZSAR by testing whether CDantiHalP generalizes to video question answering or temporal reasoning tasks, which may expand its applicability and impact.

---

> ### Author Response · Authors · 2025-11-23
> **response to reviewer 1**
>
> We sincerely thank you for the insightful comments and valuable suggestions. We have carefully addressed each point raised in the review, and additional experiments have been conducted following the recommendations. We hope our responses adequately clarify the concerns.
>
> ## **weakness 1** and **Question 1**
> 1. While the paper reports impressive results on VILA (Lin et al., 2024), LLaVA-NeXT-Video (Zhang et al., 2024), and VideoLLaMA2 (Cheng et al., 2024), **these models may no longer be at the frontier** of video understanding...
> **Add experiments on modern and stronger models** to strengthen the claim of general applicability.
>
> ### **Response:**
>
> We appreciate your valuable advice. As suggested, **we have evaluated our CDantiHalP with Qwen3-VL4B on the UCF101 dataset**. Overall, it achieves an accuracy of **80.1%**, outperforming **VILA (79.3%)**, also much better than the **baseline Tear (76.2%)**, which validates the robustness of our method.
> **More observations** are as follows:
> **Qwen3-VL demonstrates stronger multimodal understanding capabilities**. The Qwen3-VL 4B is able to apply to PromptCDs of some pairs like ('High Jump', 'Pole Vault') and ('Uneven Bars', 'Balance Beam'), which raise a relatively higher risk  (above threshold θ)  of hallucination using VILA.
>
> ## **weakness 2** and **Question 2**
> 1. **precomputing a reusable prompt library** to reduce the GPT query burden.
> 2. The current pipeline heavily relies on GPT-3.5...
> **Explore** cost-reduction strategies using **smaller open-source models** for generating and verifying prompts...,
> 3. **the verify step may potentially be done by rule-based methods** without relying on large models since it involves checking for logical contradictions. **Discussing such alternatives** would enhance practicality.
>
> ### **Response:**
>
> **Part1:**
> Thanks sincerely for your suggestion. **In our current implementation, the PromptCD and the PromptLC are already precomputed** once per action pair **and stored** for subsequent retrieval during top-2 prediction refinement of videos.
>
> **Part 2:**
> Thanks for this valuable advice. **We have replaced GPT-3.5 with the smaller open-source model Qwen3-30B** (Qwen3-30B-A3B-Instruct-2507) for prompt generation and verification. On UCF101, it **achieves an accuracy of 78.9% ( GPT-3.5 79.3%, Baseline Tear 76.2%)**. **Analysis and observations** are as follows:
> We note that in the generation of PromptCD (Figure 6), the LLM is explicitly constrained to produce deterministic PromptCD based on three predefined aspects:
> - The existence of specific objects
> - Visual attributes, such as body posture or human appearance
> - Interaction relationships, e.g., "person holds a cup"
>
> This design (1) restricts the LLM's behaviour and exploration direction; (2) ensures that similar PromptCDs can be obtained across different language models, thereby enhancing the stability and generalizability of our overall approach; (3) these also serve as effective guidance for smaller-scale models, enabling them to effectively generate discriminative PromptCDs.
> Here we provide two examples listing the PromptCDs generated by GPT-3.5 and Qwen3-30B. These two models generate similar PromptCDs, concentrating on visual attributes (same height) and the existence of a specific object (beam).
>
> Pair: "Uneven Bars", "Parallel Bars"**
> GPT-3.5: "Are the bars positioned at the same height?"
> Qwen3-30B: "Are the two bars at the same height?"
>
> Pair: 'Uneven Bars', 'Balance Beam'**
> GPT-3.5: "Is the person performing on a balance beam?"
> Qwen3-30B: "Is there a single, narrow beam?"
>
> **Part 3:**
> Thanks a lot for this valuable suggestion! **We have implemented a successful rule-based verification method** that effectively avoids performance degradation, **matching the results of the LLM-based method (79.3% on UCF101, 55.5% on HMDB51)**.
>
> **Description of this rule-based method** is as follows:
> Specifically, in generating PromptCD, we additionally require the LLM to output the target action that a 'yes' answer for PromptCD will indicate. For example, a 'yes' for PromptCD 'Is there a discuss' indicates the action to be 'throwing discuss'. We miss this information in our implementation, which is why we rely on LLM for verification. The devised simple rule is that if 'yes' is present in MLLM's response, it indicates the target action, and no/isn't/aren't/not on the contrary. This simple rule-based method significantly reduces computational overhead.

---

> ### Author Response · Authors · 2025-11-23
> **response to reviewer 1**
>
> ## **weakness 3** and **Q 3**
> - **Quantitative analysis of latency, API cost, and runtime complexity** may be missing...
> Provide detailed **efficiency analysis** ...
>
> ### **Response:**
> **We provide an analysis of the computational complexity for three main stages: prompt generation, hallucination risk assessment, and recognition refinement**. All experiments are conducted on a single A800 GPU for the UCF101 dataset, with the time consumption of API calls neglected.
>
> (1) **The generation of prompts** (PromptCDs/PromptLCs) requires **one LLM API call**. We collected 67 confused action pairs, resulting in a total of 134 API calls.
> (2) **The hallucination detection** per video requires **two rounds of MLLM inference** and **one LLM API call**. The whole risk assessment stage for all PromptCDs consumes **17 minutes** and involves **347 LLM API calls**.
> (3) **The recognition refinement** per video requires **one round of MLLM inference and one LLM API call** for verification. However, only a small portion of test videos (e.g., **20.24% of the test dataset**) necessitates refinement. This stage takes **28 minutes** and uses **766 LLM API calls**.
>
> Additionally, the **base model** (Clip/Tear) requires approximately **14 minutes** for inference. The extra computational overhead introduced by our method is **45 minutes**, which, in our view, is acceptable.
>
>
> ## **weakness 4** and **Q 4**
>
> - The experiments are limited to three standard action recognition datasets. **Evaluating on** real-world long video datasets (**e.g., ActivityNet-QA, Something-Something V2**) ...
> **Extend evaluation** beyond ZSAR by testing whether CDantiHalP generalizes **to video question answering or temporal reasoning tasks,** which may expand its applicability and impact.
>
> ### **Response:**
>
> Thank you for your suggestion to extend the evaluation. **We have conducted an experiment on the SSV2 dataset.** Since CLIP performs poorly on the SSV2 dataset (with only 0.15% top-1 and 0.94% top-5 accuracy), we developed CDantiHalP based on CAST [1]. Applied to CAST, our method **improves accuracy from 68.46% (baseline) to 68.93%,** demonstrating the generalizability of our approach. However, the improvement on SSV2 is more limited compared to the gains on UCF101 and HMDB51. This is mainly because **recognizing action pairs in SSV2 requires capturing more dynamic visual cues**. For instance, distinguishing between "Something falling like a rock" and "Dropping something onto something" depends on detecting whether an object falls heavily - a subtle distinction that poses challenges for current MLLMs.
>
> [1] Lee, Dongho, et al. "CAST: cross-attention in space and time for video action recognition." NIPS 2023

---

### Author Response · Authors · 2025-12-01
**Summary of Responses**

We sincerely thank the Area Chair and all reviewers for their time and constructive engagement with our paper.  We appreciate the reviewers' recognition of:
1. the **clever design** (R1 bpxS) and **creativity** (R3 sE3M) of our CDantiHalP
2. the **comprehensive and exhaustive experiments** (R1/2/3 bpxS/CBdX/sE3M)
3. the **portability and scalability**of our training-free framework (R1/3 bpxS/sE3M).

A **summary of our responses** is provided below:
## 1. Additional experiments and new results

In response to the reviewers’ suggestions, we conducted extensive additional experiments on **(A) New (M)LLMs**, **(B) New dataset**, **(C) New ablations**, and **(D) Computational-cost analyses** to further validate the robustness and generalizability of our CDantiHalP framework:

**(A)** When replacing GPT-3.5 with a smaller open-source LLM (Qwen3-30B), performance on UCF101 remained nearly unchanged (78.9% vs. 79.3%), while still significantly outperforming the Tear baseline (76.2%). Using a stronger MLLM (Qwen3VL-4B) further improved the accuracy to 80.1%, demonstrating the scalability of our approach.

**(B)** On the more challenging high-temporal-dependence dataset SSv2, we enhanced CAST[1]’s top-1 accuracy from 68.46% to 68.93%, confirming cross-dataset generalizability.

[1] Lee, Dongho, et al. "CAST: cross-attention in space and time for video action recognition." NIPS 2023

**(C)** We examine the performance changes with different hallucination risk level thresholds θ, observing that θ=0.8 leads to the best performance on both UCF101 and HMDB51. Furthermore, we quantify the precision of the hallucination detection (LogCHalD) as suggested, achieving a precision of 85.94% (UCF101) and 81.25% (HMDB51).

**(D)** We provide a comprehensive analysis of the computational complexity and latency of our pipeline. The three stages (prompt generation, hallucination risk assessment, and recognition refinement) require 134, 347, and 766 LLM API calls, respectively. Notably, only the latter two stages involve MLLM inference, taking 17 and 28 minutes to process the UCF101 test set. These results demonstrate that our framework is highly efficient.

## 2. Suggested model design extensions

**(A) Rule-based verification**

We replace the LLM-based verification in both the hallucination detection mechanism and the refinement stage with a rule-based method that verifies responses by directly checking for 'yes' or 'no' in them, thereby reducing computational overhead while maintaining equivalent accuracy.

**(B) Dynamic confused pair dictionary D construction**

We explore a dynamic D construction method that maintains a confusing pair pool and updates it online. It achieves 79.0% accuracy on UCF101 with a slight 0.3% decrease compared to the original 79.3%.

**(C) Adaptive hallucination risk level threshold θ selection**

We propose a dataset-adaptive θ selection mechanism that determines θ by anchoring it to the risk score of the top fixed percentage of confused pairs within each specific dataset, thus mitigating data distribution discrepancies across different datasets without any performance degradation.

## 3. Clarification for additional questions

**(A) Precompute a reusable prompt library to reduce the GPT query burden**

In our current implementation, the PromptCD and the PromptLC are indeed precomputed once per confused action pair and stored for subsequent retrieval.

**(B) The current setting is a test-aware meta-tuning of the prompt inventory**

Our method utilize test data in collecting confused pair dictionary D in an unsupervised manner, aligning with established ZSAR works like [1] which leverage test distributions for prototype adaptation. We will emphasize this in the revised paper.

[1] Universal prototype transport for zero-shot action recognition and localization. Mettes, Pascal. IJCV 2023

**(C) LogCHalD will fail when both objects can co-occur (multi-person scenes, equipment in the background)**

We clarify that our method effectively mitigates this issue by identifying such co-occurrences as a form of 'generalized hallucination.' The co-occurrence of mutually exclusive objects (e.g., a hammer and a discus) will yield a logical conflict between the responses of PromptCD and PromptLC, which is interpreted as the signal of hallucination. Furthermore, if such co-occurrences are widespread for a specific confused action pair, the corresponding PromptCD will be discarded due to the high risk level of hallucination.

**(D) How does performance degrade if the confused pair dictionary D is incomplete or noisy**

We clarify that our method is robust to incomplete or noisy D. An incomplete D results in skipping refinement for missing pairs (falling back to the baseline), while a noisy D contains extraneous entries that are never retrieved. In both cases, there is no performance degradation or wasted computational resources.

---

### Meta-Review · Area_Chair_GrQm · 2026-01-06

**Summary:**

This paper proposes a training-free and plug-and-play CDantiHalP framework for the Zero-Shot Action Recognition task. The submission was reviewed by three experts in the field, with initial scores 6/4/6, respectively.

All reviewers raised concerns about the potential inefficiency since the CDantiHalP requires multiple MLLM/LLM API calls. More importantly, Reviewer CBdX and sE3M raised a key question that the construction of the confused-pair dictionary requires accessing the test dataset, which undermines the zero-shot generalizability of CDantiHalP. Also, Reviewer sE3M raised concern regarding the robustness of the framework in the case of an incomplete and noisy confused-pair dictionary. Besides, Reviewer bpxS raised a major concern regarding the limited evaluation that lacks experiments on SOTA Video LLMs and standard action recognition datasets. While the AC has reviewed the authors' rebuttal, the rebuttal contains few quantitative results, and certain concerns remain unresolved. The authors are encouraged to address these points by incorporating the reviewers’ suggestions into their revisions to further strengthen the paper.

Given ICLR’s exceptionally competitive acceptance rate this year, the AC regrets to recommend rejection.

**Reviewer Concerns:**

Addressed concerns:

1. Potential inefficiency of the CDantiHalP framework due to multiple MLLM/LLM api calls (All reviewers).

2. The effectiveness of the CDantiHalP using smaller LLM backbones instead of using GPT-3.5 (bpxS).

3. The potential limitation and inflexibility of the static risk threshold (CBdX).

4. Can the confused-pair dictionary be built dynamically during inference (sE3M).

Unaddressed concerns:

1. Additional experiments on SOTA Video LLMs and complex video understanding benchmarks. (bpxS).

2. The limited generalizability of the confused-pair dictionary (CBdX and sE3M).

3. LogCHalD may fail in the case of object co-occurrence (CBdX).

The reasons these concerns were not well addressed are presented in the "Reviewer Scores" section.

**Reviewer Scores:**

The paper received three reviews, with scores 6/4/6 respectively.

- Reviewer bpxS (6): The score is unlikely to be changed. The reviewer required more experiments on SOTA Video LLMs (e.g., Qwen3-VL) and complex video understanding benchmarks (e.g, ActivityNet-QA). The AC checked and found that, although the rebuttal presented new results on Qwen3-VL, it remains unclear whether the performance gains come from the CDantiHalP or the stronger temporal understanding capabilities of Qwen3-VL. In addition, the new results on the SSV2 dataset indicated limited performance gains of CDantiHalP (68.46->68.93) when applied to complex video understanding datasets.

- Reviewer CBdX (4): The score is unlikely to be changed. The reviewer raised a key concern regarding the confused-pair dictionary. The proposed CDantiHalP requires access to the test dataset for constructing the dictionary, undermining the zero-shot generalizability of CDantiHalP when applied to new data distributions. According to the rebuttal provided by the authors, it is still found unclear whether the confused-pair dictionary requires recollection when applied to a new testset. Additionally, Reviewer CBdX noted that the CDantiHalP uses a static threshold based on empirical settings, while the AC found that the authors didn't respond to this concern.

Although the authors provided empirical analysis regarding the potential limitation of LogCHalD in the case of object co-occurrence, they didn't present more experimental results on more complex video understanding benchmarks (e.g., ActivityNet-QA), instead of the standard action recognition datasets, while lacking qualitative analysis, which makes the claims less convincing.

- Reviewer sE3M (6): The score is unlikely to be changed. The reviewer also raised a key concern regarding the zero-shot generalizability of the confused-pair dictionary. In addition, another key concern regarding the robustness of the confused-pair dictionary. However, the author didn't present sufficient experimental results in the case of the incomplete and noisy confused-pair dictionary by decreasing the number of sampled test videos during construction, and the concern was thus not fully addressed.

---

### Decision · Program_Chairs · 2026-01-26

Reject